# Experimental certification of millions of genuinely entangled atoms in a solid

Florian Fröwis [1], Peter C. Strassmann [1], Alexey Tiranov[1], Corentin Gut[1,2], Jonathan Lavoie[1,3], Nicolas Brunner[1], Félix Bussières[1], Mikael Afzelius[1] & Nicolas Gisin[1]

Quantum theory predicts that entanglement can also persist in macroscopic physical systems, albeit difficulties to demonstrate it experimentally remain. Recently, significant progress has been achieved and genuine entanglement between up to 2900 atoms was reported. Here, we demonstrate 16 million genuinely entangled atoms in a solid-state quantum memory prepared by the heralded absorption of a single photon. We develop an entanglement witness for quantifying the number of genuinely entangled particles based on the collective effect of directed emission combined with the non-classical nature of the emitted light. The method is applicable to a wide range of physical systems and is effective even in situations with significant losses. Our results clarify the role of multipartite entanglement in ensemble-based quantum memories and demonstrate the accessibility to certain classes of multipartite entanglement with limited experimental control.

[1] Groupe de Physique Appliquée, Université de Genève, CH-1211 Genève, Switzerland. [2] Present address: Institut für Theoretiche Physik, Leibniz Universität Hannover, Hannover, Germany. [3] Present address: Department of Physics and Oregon Center for Optical Molecular & Quantum Science, University of Oregon, Eugene, OR 97403, USA. Peter C. Strassmann and Alexey Tiranov contributed equally to this work  Correspondence and requests for materials should be addressed to F.F. (email: florian.froewis@unige.ch)

A clear picture of large-scale entanglement with its complex structure is so far not developed. It is, however, important to understand the role of different facets of multipartite entanglement in nature and in technical applications[1, 2]. For example, the so-called Schrödinger cat states[3] are fundamentally different from a single-photon coherently absorbed by a large atomic ensemble; even though both are instances of multipartite entanglement (ref. [4], chapter 16.5). The theoretical study of large-scale entanglement has to be followed by an experimental demonstration, which consists of two basic steps: the preparation of an entangled system and a subsequent appropriate measurement verifying the presence of entanglement. In the context of entanglement in large systems, the preparation of entanglement is generally much simpler than its verification. For example, single-particle measurements are often not possible and collective measurements are typically restricted to certain types and are of finite resolution. These limitations call for new witnesses that allow one to certify entanglement based on accessible measurement data.

The concept of entanglement depth[5] was shown to be meaningful for and applicable to large quantum systems. It is defined as the smallest number of genuinely entangled particles that is compatible with the measured data. This allows one to witness at least one subgroup of genuinely entangled particles in a state-independent and scalable way. Large entanglement depth was successfully demonstrated with so-called spin-squeezed and oversqueezed states by measuring first and second moments of collective spin operators[6–9]; lately up of 680 atoms[10]. Similar ideas were realized for photonic systems[11, 12]. Recently, a witness was proposed that is designed for the W state, which is a coherent superposition of a single excitation shared by many atoms[13]. Based on this witness, an entanglement depth of around 2900 was measured[14]. However, these witnesses do not detect entanglement when the vacuum component of the state is dominant[13], even though the W state is known to be quite robust against various sources of noise, in particular, against loss of particles and

excitation[15]. Hence, much larger values for the entanglement depth could be expected.

In this paper, we present theoretical methods and experimental data that verify a large entanglement depth in a solid-state quantum memory. A rare-earth-ion-doped crystal spectrally shaped to an atomic frequency comb (AFC) is used to absorb and re-emit light at the single-photon level[16–19], where at least 40 billion atoms collectively interact with the optical field. Using the measured photon number statistics of the re-emitted light we collect partial information about the quantum state of the atomic ensemble before emission. Then, we show that certain combinations of re-emission probabilities for one and two photons imply entanglement between a large number of atoms. With the measured data from our solid-state quantum memory we demonstrate inseparable groups of entangled particles containing at least 16 million atoms.

## Results

**Intuition behind detecting many-atom entanglement.** Before discussing the experiment, we give an intuitive explanation for the appearance of large entanglement depth when a large atomic ensemble coherently interacts with a single photon (Fig. 1a). Suppose that $N$ two-level atoms ($|g\rangle$ and $|e\rangle$ denote ground and excited state, respectively), couple to a light field. The quantized interaction in the dipole approximation is described by[20]

$$H_{\text{int}} = \sum_{j,\mathbf{k}} e^{-i\mathbf{k}\cdot\mathbf{r}_j} a_{\mathbf{k}} \sigma_+^{(j)} + e^{i\mathbf{k}\cdot\mathbf{r}_j} a_{\mathbf{k}}^\dagger \sigma_-^{(j)}, \quad (1)$$

that is, a single photon with wave vector $\mathbf{k}$ is annihilated by exciting atom $j$ via $\sigma_+|g\rangle = |e\rangle$ and vice versa. The phase is given by the scalar product between $\mathbf{k}$ and the position $\mathbf{r}_j$ of the atom. When an incoming light field is absorbed via interaction (1), the imprinted phase relation between the atoms serves as a memory for the direction and the energy of the absorbed photons. Without this information, a spontaneous, directed re-emission is

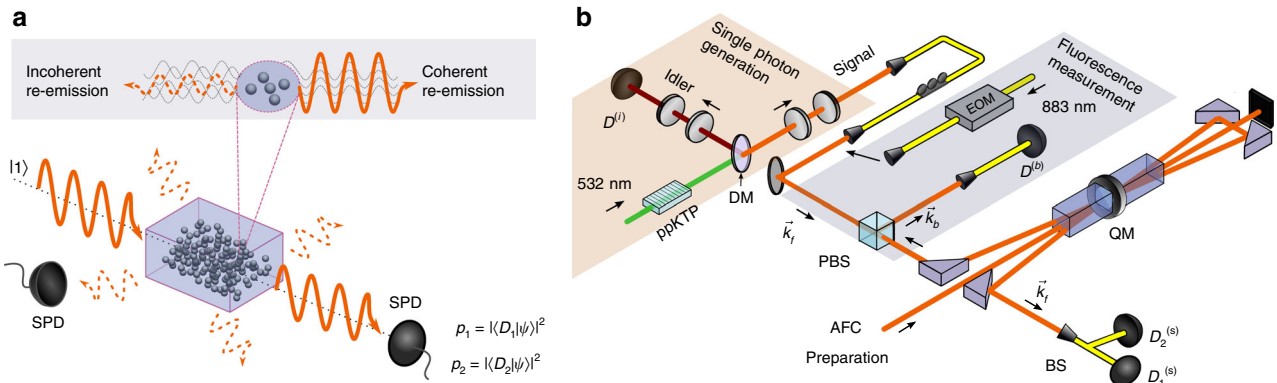

**Fig. 1** Basic intuition and experimental setup. **a** When atoms spontaneously emit photons, phase coherence between the atoms leads to constructive interference and enhanced emission probability in a certain direction, measured by a single-photon detector (SPD). Emission in any other direction is incoherent and hence not enhanced. If this phase coherence is generated by absorbing a single photon, the atoms are necessarily entangled. **b** The experiment consists of the heralded single-photon source, the quantum memory (QM), the detection system in the forward mode $\mathbf{k}_f$ and the fluorescence measurement in the backward mode $\mathbf{k}_b$ of the QM. The source is based on a spontaneous parametric down conversion process. A periodically poled KTP (ppKTP) waveguide is pumped by a monochromatic laser at 532 nm wavelength which leads to the generation of photon pairs. They contain signal (idler) photons at 883 nm (1338 nm) wavelength spatially separated by a dichroic mirror (DM). The detection of the idler photon ($D^{(i)}$) heralds the presence of the signal photon in a well defined spectral, temporal and polarization mode. The heralded single photon is absorbed by the quantum memory which is based on two $Nd^{3+}$:$Y_2SiO_5$ crystals. A double-pass configuration is used to enhance the absorption process. To estimate $p_1$ and $p_2$, the one-photon and two-photon probabilities from the re-emission process are measured in the forward direction, $\mathbf{k}_f$, using a fiber-based 50/50 beamsplitter (BS) and two SPDs $D_1^{(s)}$ and $D_2^{(s)}$. In order to measure the number of atoms $N$, the single-photon source is replaced by a bright coherent state created using an electro-optical modulator (EOM). This increases re-emission intensities in forward and backward direction. The backward direction is measured by placing a polarization beamsplitter (PBS) in the input mode of the memory and using a SPD $D^{(b)}$

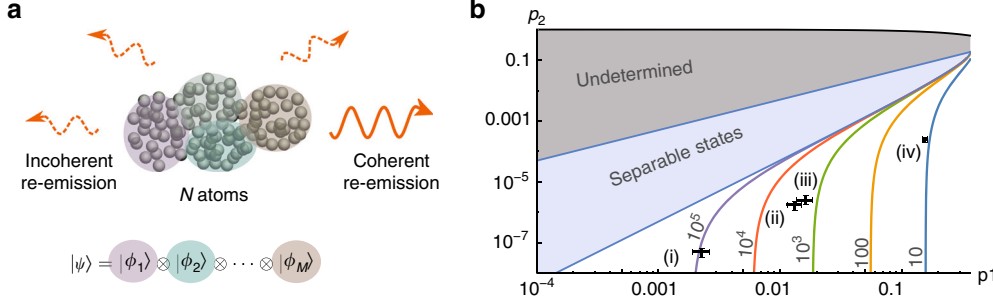

**Fig. 2** Illustration of the basic ansatz and results of the entanglement witness. **a** The colored areas in the ensemble are genuinely entangled, while no entanglement is present between the groups. **b** The minimization of the two-excitation probability $p_2$ for given single-excitation probability $p_1$ and number of separable groups $M$ leads to lower bounds which are independent of $N$ if $N \gg 1$. The central region in the plot is spanned by separable states (i.e., $M = N$). Entanglement is required to reach smaller $p_2$ while keeping $p_1$ constant. The number next to a colored line is the maximal $M$ that is compatible with data points on this line. This $M$ is then used to bound the entanglement depth $K = N/M$. The four black crosses are data points from the experiment including one standard deviation, where different levels of inefficiencies are taken into account. Data point (i) is directly inferred from the raw data. Data point (ii) is obtained from (i) by removing the effect of finite detector efficiencies. Data points (iii) and (iv) are more speculative as these points remove the effect of the re-emission efficiency (for (iii) and (iv)) and the re-phasing efficiency (for (iv)). A maximization of $p_2$ given $p_1$ and $M$ would be necessary to make statements about the gray top zone (undetermined)

not possible. In other words, phase coherence between the atoms is necessary in order to a have well-controlled re-emission direction[21, 22]. Now, depending on the nature of the absorbed light, this coherence implies entanglement between the atoms or not. On the one hand, the absorption of a coherent state leads to a coherent atomic state, which is unentangled (ref. [4], chapter 16.7). On the other hand, if a single photon $|1\rangle$ is absorbed, the quantization of the field leads to a W state (or Dicke state with a single excitation) of the atomic state (ref. [4], chapter 16.5)

$$|1\rangle \rightarrow |D_1\rangle \propto \sum_j e^{-i\mathbf{k}\cdot\mathbf{r}_j}|g\ldots ge_jg\ldots g\rangle. \tag{2}$$

Then, the ensemble is genuinely multipartite entangled[15]. These examples suggest a generic relation between directed emission, single-photon character of the emitted light and large entangled groups.

In our experiment, we use a neodymium-based solid-state quantum memory operating at a total read–write efficiency of 7% (Fig. 1b). This memory was demonstrated to be capable of storing different types of photonic states and preserving state properties such as the single-photon character[17, 19, 23–25]. A heralded single photon is produced via spontaneous parametric down conversion (SPDC)[26] and coupled to the atomic ensemble, which was prepared in the ground state $|D_0\rangle = |g\rangle^{\otimes N}$. After a 50 ns delay time, the coherent excitation is spontaneously re-emitted in forward direction and detected. In practice, this optical state is not exactly a single photon. Due to losses at different levels, the state contains a large vacuum component. Also higher photon components are present. However, since directed emission and non-classical photon number statistics are largely preserved, entanglement between large groups of atoms is expected.

**Derivation of the entanglement depth witness**. In order to certify this entanglement, we develop the following entanglement witness (see Methods section for details). Consider a pure state that is subdivided into a product of $M$ groups

$$|\psi\rangle = |\phi_1\rangle \otimes \ldots \otimes |\phi_M\rangle, \tag{3}$$

where the $|\phi_i\rangle$ are arbitrary. Phase coherence between the groups implies that each group has to carry some excitation. This necessarily amounts to an emission spectrum that also contains multi-photon components.

To be more specific, we consider the probabilities of the atoms emitting one and two photons, $p_1$ and $p_2$, respectively. In the low-excitation limit (see Methods section), these probabilities correspond to $p_1 = |\langle D_1|\psi\rangle|^2$ and $p_2 = |\langle D_2|\psi\rangle|^2$, where

$$|D_2\rangle \propto \sum_{j<l} e^{-i\mathbf{k}\cdot(\mathbf{r}_j+\mathbf{r}_l)}|g\ldots ge_jg\ldots ge_lg\ldots g\rangle, \tag{4}$$

that is, the phase-coherent superposition of two excitatons. As shown in the Methods section, it is possible to find the minimal $p_2$ for a given $p_1$ within the class (3) with fixed $M$. By varying $p_1$ and $M$ one finds a lower bound on $p_2$ as a function of $p_1$ and $M$. Given the linearity of $p_1$ and $p_2$ when mixing states like in Eq. (3) (with arbitrary grouping but lower-bounded $M$), the extension of the bound to mixed states is straightforward. Examples of such lower bounds are shown in Fig. 2. Note that the bounds are independent of $N$ if $N \gg 1$. Comparing the lower bounds with experimental data in turn gives an upper bound on $M$ and, by additionally measuring $N$, a lower bound on the entanglement depth, which simply reads $K = N/M$.

**Experimental realization and measured data**. Experimentally, $p_1$ is obtained from the probability to measure a single re-emitted photon in the forward mode ($\mathbf{k}_f$ in Fig. 1b) at a predetermined time, which we herald by the detection of the idler photon at the source. The value of $p_2$ corresponds to the two-photon statistics of the re-emitted light in the $\mathbf{k}_f$ mode. It is inferred from the measured autocorrelation function $g_{\text{ss}|i}^{(2)} = 2p_2/p_1^2$ and $p_1$. The identification of photonic Fock states with atomic Dicke states is possible because the memory is initially prepared in the ground state and because of the photon number statistics of the source (Methods section). From the raw data, we find $p_1 = 2.3(3) \times 10^{-3}$ and $p_2 = 5(2) \times 10^{-8}$. The relatively small value of $p_1$ is a product of the efficiencies of the source, the memory and detectors. The partial subtraction of these losses leads to different values of the effective $p_1$ (Fig. 2, Table 1, and Methods): (i) raw data; (ii) subtraction of detector noise, that is, the actual value for the emitted light; (iii) subtraction of the re-emission inefficiency, that is, the single-excitation component of the atomic ensemble just before emission; (iv) subtraction of phase noise during storage, that is, overlap with the $|D_1\rangle$ state right after absorption. The values of $p_2$ directly follow using the measured autocorrelation function $g_{\text{ss}|i}^{(2)} = 0.020(3)$, which is—under conservative assumptions—not affected by this modeling.

**Table 1 Results for entanglement depth $K$**

| Level of modeling | $p_1$ | $K$ | $K - 3\sigma$ |
|---|---|---|---|
| (i) Raw data | 0.0023(3) | $4.76 \times 10^5$ | $7.54 \times 10^4$ |
| (ii) After re-mission | 0.013(2) | $1.64 \times 10^7$ | $3.72 \times 10^6$ |
| (iii) Before re-mission | 0.016(2) | $2.46 \times 10^7$ | $5.24 \times 10^6$ |
| (iv) After absorption | 0.16(1) | $3.23 \times 10^9$ | $2.09 \times 10^9$ |

Depending on the level of modeling the inefficiencies of the experimental setup, different values for $p_1$ and hence for $K$ are obtained (cf. Fig. 2b). By sampling $p_1$, $p_2$, and $N$ around the measured values within the estimated uncertainties, we calculate the expected entanglement depth $K$. The values in the last columns are lower bounds on $K$ with confidence $3\sigma = 99.7\%$

A key element in the experiment is the high-precision measurement of $N$. For this, the relation between ensemble size and directionality of the re-emitted light is exploited. The ratio of the coherent emission in the forward direction and the incoherent emission in the backward direction is a lower bound on the number of resonant atoms[22]. Since incoherent emission from single photons is much lower than detector dark counts, the single-photon source is replaced by a bright coherent state for this measurement (Fig. 1b and Methods section) and we find $N \geq 4.0$ $(1) \times 10^{10}$. The resulting $K$ depends on the level of modeling (i) to (iv) as mentioned before (Table 1). Our data analysis illustrates how decoherence and noise reduces the certifiable entanglement depth. Immediately after absorption of the single photon (iv) we have an entanglement depth of about $10^9$; this reduces to about $10^7$ just before and immediately after re-emission (iii) and (ii), respectively, while when taking into account all losses and detector inefficiencies the certifiable entanglement depth drops to about $10^5$ (i). In our opinion, a conservative but reasonable number of the certified entanglement depth is $10^7$. Indeed, on the one side the entanglement depth of $10^9$ in (iv) relies more on our theoretical model than on our data, while on the other side the $10^5$ value in (i) takes into account well-understood losses that are not part of the physical phenomenon we aim to certify.

## Discussion

This work demonstrates that large entanglement depth is experimentally certifiable even with atomic ensembles beyond $10^{10}$ atoms and low detection and re-emission efficiencies. We prove that entanglement between many atoms is necessary for the functioning of quantum memories that are based on collective emission, because the combination of directed emission (i.e., high-memory efficiency) and preservation of the single-photon character imply large entanglement depth.

Our results further illustrate the fundamental difference between various manifestations of large entanglement. The scales at which we observe entanglement depth seem to be completely out of reach for other types of large entanglement, such as Schrödinger-cat states[2, 27].

As detailed in the Methods section, our reasoning is based on two steps. First, a model-independent witness for entanglement depth is derived, which only depends on the overlap of the atomic state with $|D_1\rangle$ and $|D_2\rangle$ as well as the total number of atoms, $N$. In the experiment, we measure the probabilities $p_1$ and $p_2$ for one and two photons, respectively, emitted from the atomic ensemble. The second step consists in identifying $p_1$ and $p_2$ with the probabilities of the atomic ensemble being in the $|D_1\rangle$ and $|D_2\rangle$ state before the re-emission, respectively. This step as well as the measurement of $N$ are based on some assumptions regarding the atomic ensemble, the single-photon source and the light-matter interaction, Eq. (1). In addition, our claimed entanglement depth in the order of $10^7$ takes finite detector efficiencies into account. We emphasis, however, that these assumptions have been thoroughly tested in the classical and quantum regime in many

previous experiments. Further note that the entanglement depth is generated by a probabilistic but heralded source. Hence no post-selection has been made in our experiment.

We report lower bounds on the minimal number of genuinely entangled atoms, which should not be confused with quantifying entanglement with an entanglement measure. Indeed, the nature of the target state, the W state $|D_1\rangle$, and the experimental challenges suggest that only a small amount of entanglement is present in the crystal during the storage.

We note that entanglement between many large groups of atoms in a solid is demonstrated in a parallel submission by Zarkeshian et al., where the coherence between these groups is revealed by analyzing the temporal profile of the re-emission in the forward direction.

## Methods

**Heralded single-photon source.** The single photon used to prepare the entangled state of the atomic ensemble is generated using SPDC. A 2 mW monochromatic continuous-wave 532 nm laser pumps a periodically poled potassium titanyl phosphate waveguide to generate the signal and idler photons at 883 and 1338 nm, respectively. The two down-converted photons are energy-time entangled. The narrow spectral filtering of the signal (idler) photon is performed using a Fabry–Perot cavity with a linewidth of 600 MHz (240 MHz)[26]. The detection of the idler photon heralds the presence of a signal photon in a well defined spectral, temporal and polarization mode. The heralded single photon is then absorbed in the quantum memory. The zero-time second-order autocorrelation of the heralded single photon before storage in the quantum memory was measured to be $g_{ssli} = 0.0055(2)$ when using a 1.2 ns coincidence window. The idler mode is detected by an InGaAs/InP single-photon detector ID220 from ID Quantique (20% detection efficiency), while the signal mode is analyzed using silicon avalanche photodiodes from Perkin Elmer (30% detection efficiency).

**Solid-state quantum memory.** The single-photon storage is performed using a broadband and polarization-preserving quantum memory[19] realized by placing two 5.8 mm-long 75 ppm $Nd^{3+}$:$Y_2SiO_5$ crystals around a 2 mm-thick half-wave plate. An AFC is prepared on the center of the $Nd^{3+}$ ions transition at 883 nm (absorption line $^4I_{9/2} \rightarrow {}^4F_{3/2}$) by optical pumping. Using optical path consisting of acousto-optic and phase modulators we create a 600 MHz comb with a spacing of 20 MHz between the absorption peaks[24]. The resulting optical depth of the absorption peaks is $d = 2.0 \pm 0.1$. This value was doubled using a double-pass propagation through the crystals (Fig. 1b). The overall single-photon efficiency of the AFC quantum memory is 7(1)% with a 50 ns storage time. The absorption probability of one photon by the crystal was estimated to be 82(1)%, which was obtained from the probability for the single photon to be transmitted.

**One-photon and two-photon probabilities.** The one-photon and two-photon probabilities in the forward mode are obtained as follows. First, the transmission probability along the path from the photon pair source to the single-photon detectors was carefully estimated using heralded single photons. The overall transmission consists of (i) the heralding probability of the single photon before the quantum memory 19(1)%; (ii) the overall quantum memory efficiency 7(1)%; and (iii) the detection efficiency including the transmission of the system 17(1)%. From this we found that the total probability to detect a single-photon re-emitted from the crystal is $p_1 = 2.3(3) \times 10^{-3}$. Then, the probability $p_2$ can be estimated from a measurement of the zero-time second-order autocorrelation function $g^{(2)}_{ss|i} = 2p_2/p_1^2$. We measured $g^{(2)}_{ss|i} = 0.020(3)$. This value is higher than the one before the storage, which is due to spurious noise coming from the photon pair source[28]. The probability to detect two photons is estimated to be $p_2 = 5(2) \times 10^{-8}$.

To connect the experimental observation of one and two photons with $p_1 = \langle D_1|\rho|D_1\rangle$ and $p_2 = \langle D_2|\rho|D_2\rangle$, respectively, some details have to be clarified. First, we note that the actual coupling between light and atoms is not uniform as in Eq. (1) due to position dependent field intensities and the inhomogeneous broadening of the atomic ensemble. However, it is possible to mathematically replace this by an ideal, uniform coupling with a reduced ensemble size[29]. We expect that replacing an ensemble by a smaller one only lowers the bounds on entanglement depth. Since $N \gg 1$ and the weak coupling between the field and a single atom, the dynamics from Eq. (1) are well approximated by a first-order expansion of the Holstein–Primakoff transform[30], that is, the linear regime

$$U^\dagger a_{\mathbf{k}} U = \sqrt{\eta} N^{-1/2} S_-^{\mathbf{k}} + \sqrt{1 - \eta} a_{\mathbf{k}}, \quad (5)$$

where $\eta$ is the transfer efficiency and $S_\pm^{\mathbf{k}} = \sum_{j=1}^N e^{\mp i \mathbf{r}_j \cdot \mathbf{k}} \sigma_\pm^{(j)}$ are the creation and annihilation operators for a collective atomic excitation (up to the normalization factor $N^{-1/2}$ [31]. All formulas in this paper are based on this approximation and the next-order correction $O(1/N)$ is omitted.

Second, the states $|D_1\rangle$, Eq. (2), and $|D_2\rangle$, Eq. (4), are not the only ones that give rise to the emission of one and two photons, respectively. Let us introduce the canonical basis $\{|j, m, \alpha\rangle_\mathbf{k}\}_{j,m,\alpha}$ for the angular momentum operators $S_z = \frac{1}{2}\left[S_+^\mathbf{k}, S_-^\mathbf{k}\right]$ and $S_\mathbf{k}^2 = S_z^2 + \frac{1}{2}\{S_+^\mathbf{k}, S_-^\mathbf{k}\}$, where $S_\mathbf{k}^2 |j, m, \alpha\rangle_\mathbf{k} = j(j+1)|j, m, \alpha\rangle_\mathbf{k}$ and $S_z|j, m, \alpha\rangle_\mathbf{k} = m|j, m, \alpha\rangle_\mathbf{k}$. The third quantum number $\alpha$ labels the degeneracies for asymmetric states. For $\eta = 1$, any atomic state $|j, -j+l, \alpha\rangle_\mathbf{k}$ with $N/2 - j = O(1)$ (low-excitation limit) and $\mathbf{k}$ the forward mode transforms via Eq. (5) to the photonic Fock state $|l\rangle$. Now, assuming that the single-photon source is the only source of coherent excitation, the population of the subspaces $N/2 + m$ is strongly decaying with $m$, such that the overlap of the atomic state before re-emission with $|D_1\rangle \equiv |N/2, -N/2 + 1, 1\rangle_\mathbf{k}$ is much larger than with the entire subspace spanned by $\{|j, -j+1, \alpha\rangle_\mathbf{k}\}_{j<N/2}$ (i.e., the nonsymmetric subspace emitting a single photon). Using the $g_{\mathrm{ss|i}}^{(2)}$ directly measured at the source, it can be estimated that corrections taking the nonsymmetric subspace into account are much smaller than the uncertainty of the measured $p_1$. A similar argument applies to the two-photon emission.

Finally, the memory preparation ideally sets the atomic ensemble to the ground state. In practice, we estimate that roughly $10^{-5} \times N$ atoms are at the end of the preparation phase in the excited state without any phase coherence between them. In the linear regime, these excitations simply drop out from all calculations and can hence be safely ignored.

**Number of atoms in the atomic ensemble.** Another parameter is the number of atoms $N$ participating to the collective atomic mode, which is estimated with a separate measurement. The ratio between coherent (signal) and incoherent (noise) emission from the atomic ensemble was used to estimate the number of atoms coupled to the optical mode. A simple model for this signal-to-noise ratio (SNR) was developed, where the number of atoms $N$ is a free parameter. By independent characterization of the remaining other parameters and by measuring the SNR, we obtain and estimate of $N$. Intuitively, this is based on the fact that the re-emission from the atomic ensemble is enhanced in the spatial mode of the incident single photon, due to the constructive interference between all the atoms which have collectively absorbed the single photon. This probability ideally equals to $N p_\mathrm{s}$, where $p_\mathrm{s}$ is the probability for spontaneous emission of a single atom[22]. In any other mode, including the backward mode, there is no collective enhancement, and the probability of an incoherent re-emission is just $p_\mathrm{s}$. Hence, the SNR is given by $N$ in the ideal case where no other source of noise is present.

In principle, one could measure the SNR from the probability to detect the heralded single photon in the backward mode. In practice, this cannot be done because $N \sim 10^{10}$. The incoherent re-emission probability is extremely small and is therefore lost in the noise due to detector dark counts and spurious light. To overcome this limitation, strong coherent state pulses with mean photon number $|\alpha|^2$ up to $10^6$ were used instead to estimate the SNR. This value is still much lower that the total number of atoms which keeps the interaction in linear regime. In this case the noise becomes less important and the true incoherent re-emission can be measured. To detect it with a low noise level we used a Picoquant silicon avalanche photodiode detector with 35% efficiency and 4 Hz dark count rate.

To perform the SNR measurement the forward $\mathbf{k}_\mathrm{f}$ and the backward $\mathbf{k}_\mathrm{b}$ spatial modes were used to measure signal and noise, respectively (Fig. 1b). For each mode spatial filtering was performed using single-mode fibers. This allowed us to confirm that modes in both directions (forward $\mathbf{k}_\mathrm{f}$ and backward $\mathbf{k}_\mathrm{b}$) are probing the same volume of the QM. For this the light was sent in both directions and the coupling was aligned simultaneously for both couplers after the PBS, as shown in Fig. 1b. Furthermore, the incoherent re-emission from the strong coherent state pulses was detected simultaneously in both modes. By applying corrections for diverse optical losses the ratio between the intensities in both modes was found to be very close to 1 as expected (Supplementary Note 1). This confirms that the forward $\mathbf{k}_\mathrm{f}$ and backward $\mathbf{k}_\mathrm{b}$ modes correspond to the coherent and incoherent re-emission modes defined by our model. Note that any systematic error that leads to an underestimated signal or to overestimate noise only reduces the inferred $N$ and hence leads to an underestimated entanglement depth. For example, scattered photons from the $\mathbf{k}_\mathrm{f}$ mode that could have been mistakenly collected in the $\mathbf{k}_\mathrm{b}$ mode would increase the noise and hence would lead to an underestimated $N$. We are not aware of any systematic error in our experiment that would let us overestimate $N$.

Using a coherent state pulse with a mean number of photons equal to $|\alpha|^2$, the signal is proportional to $\eta|\alpha|^2$, where $\eta$ is the re-phasing efficiency of the quantum memory. The incoherent re-emission in the backward mode is proportional to $|\alpha|^2/N + \delta$, where $\delta$ is a noise probability. Hence, the SNR is given by $\eta|\alpha|^2/(|\alpha|^2/N + \delta)$, from which $N$ can be obtained (Supplementary Note 1). From this, the number of atoms was found to be $N = 4.0(1) \times 10^{10}$. An estimate for the number of atoms obtained by considering the doping concentration, the length of the crytals and the size of the optical mode roughly gives $3 \times 10^{11}$, which confirms at least the order of magnitude. Note, however, that the latter method comes with much larger uncertainties and therefore we rely only on the first number.

**Ansatz for M-separability.** We now give some details for the derivation of a lower bound of $p_2$ given $p_1$ and the ansatz state (3). We start with the ansatz that for every pure state decomposition of an atomic state the pure states are separable between at

least $M$ groups (where each group is genuinely multipartite) and there exists at least one pure state in every decomposition that consists of exactly $M$-separable groups. Such a state is called $M$-separable. In principle, the sizes of the groups are independent from each other as long as the total number of atoms is conserved. However, we fix the group size $K$ to be constant, that is, $MK = N$ for the following reason. Our final goal are bounds on numbers of entangled atoms. This is a "min-max" problem. For every possible state the entanglement depth is the size of the largest entangled group in the state. By varying the state, our task is to find a state such that this largest group is minimized. From this, it follows that it is best to have an equal size for all groups in order to avoid few very large groups. Clearly, if we fix $N$ and $M$, $K$ does not have to be an integer. So, generally, one has to reduce the size of one group such that $(M-1)K + K' = N$. However, we will consider many groups such that the size of a single group is in the order of or smaller than the uncertainty of $N$. Hence a detailed analysis with $K' < K$ is not necessary.

Since we are concerned with at most two excitations in total, it is sufficient to work with pure states of the form

$$|\psi\rangle = \bigotimes_{i=1}^{M} (a_i|d_0\rangle + b_i|d_1\rangle + c_i|d_2\rangle), \qquad (6)$$

where $|d_k\rangle$ here refers to Dicke states within one group, that is, symmetric superposition of $k$ excitations. With this ansatz, the probabilities read

$$p_1 = \frac{|A|^2}{M}\left|\sum_i \frac{b_i}{a_i}\right|^2 \qquad (7)$$

and

$$p_2 = \frac{|A|^2}{M^2\left(1 - \frac{1}{N}\right)}\left|\sqrt{2}\sum_{i<j}\frac{b_i b_j}{a_i a_j} + \sqrt{1 - \frac{1}{K}}\sum_i \frac{c_i}{a_i}\right|^2 \qquad (8)$$

where $A = \Pi_{i=1}^M a_i$. In the following, we ignore the corrections $1/N$ and $1/K$. While the factor $(1 - 1/N)^{-1}$ is arguably negligible, dropping $\sqrt{1 - 1/K}$ only lowers $p_2$ in the relevant regime (i.e., the interval $I$ discussed later).

Formally, the task is now to minimize $p_2$ for given $p_1$, $M$ over the parameters of the ansatz state (6), that is,

$$p_2^{\min}(p_1, M) = \min_{\psi: p_1 = \mathrm{const}} p_2. \qquad (9)$$

When this is done for all $p_1$, one has to find the minimum of all convex combinations for a bound on mixed states. In our case, it will turn out that the lower bounds for pure states are already convex implying that they are also valid for mixed states. Then, one can invert the results and determine the maximal $M$ that is compatible with a given pair $(p_1, p_2)$. This bounds the $M$-separability of the atomic state generated in the experiment.

Since the state (6) could contain further elements not contributing to $p_1$ and $p_2$, one has in general that $|a_i|^2 + |b_i|^2 + |c_i|^2 \leq 1$ for all $i$. Thus the minimization (9) has to be done over $3M$ complex parameters. One easily shows that the complexity reduces to $2M$ real parameters as the optimal state has $a_i \geq 0$, $b_i \in \mathbb{R}$ and $c_i = -\sqrt{1 - a_i^2 - b_i^2}$ (Supplementary Note 2).

**Lagrange multiplier.** To solve Eq. (9), we use the Lagrange multiplier method, which is suitable for constrained minimization problems. In our case, we consider

$$f(\{a_i, b_i\}_i) = f_2(\{a_i, b_i\}_i) + \lambda[f_1(\{a_i, b_i\}_i) - C] \qquad (10)$$

with $f_1 = \sqrt{M p_1}$ and $f_2 = \sqrt{M^2 p_2}$. The formulas for the partial derivatives are given in the Supplementary Note 3. The results are quartic equations with four solutions for every group $i > 1$ that depend on the parameters of the first group $a_1 \equiv a$ and $b_1 \equiv b$ for all $i$.

Note that the first solution (called the symmetric solution in the following) implies that $a_i = a$ and $b_i = b$. The probabilities in this case read

$$p_1^{\mathrm{sym}} = M a^{2M-2} b^2 \qquad (11)$$

and

$$p_2^{\mathrm{sym}} = a^{2M}\left(\frac{1}{\sqrt{2}}(M-1)\frac{b^2}{a^2} + \frac{c}{a}\right)^2, \qquad (12)$$

with $c = -\sqrt{1 - a^2 - b^2}$.

Fixing $a$, $b$ of the first group determines the four possible solutions for every other group. A priori, the unknowns $(a, b, \lambda)$ can be found by solving the remaining equations $\partial f/\partial a = 0$, $\partial f/\partial b = 0$ and $\partial f/\partial \lambda = 0$. Given the complexity of the equations, this is analytically not possible. Alternatively, one chooses for each group $i = 2, \ldots, M$ one out of four solutions, and solve Eq. (9) numerically for $(a, b)$. As a result, we find many local extrema, from which one has to choose the global minimum. Hence, we reduced the minimization problem to a finite set of possibilities. The problem is the large number of solutions. Due to symmetry, it is sufficient to determine the number of groups, $m_j$, that are chosen to take solution $j = 1, \ldots, 4$

with the constraints $m_1 > 0$ and $\sum_j m_j = M$ (Supplementary Note 4). In the following, we call a certain choice $C = (m_1, m_2, m_3, m_4)$ a configuration. The number of configurations scales as $O(M^3)$.

It turns out that most of the configurations are not relevant for the following reason. For all $M$, there exist states such that $p_1 > 0$ and $p_2 = 0$. Two analytic examples are (i) $(a_1, b_1) = (0, 1)$ and $(a_{i>1}, b_{i>1}) = (1, 0)$, resulting in $p_1^{\lim 1} = 1/M$ and (ii) a symmetric state where one maximizes $p_1$ given that Eq. (12) vanishes. The solution $p_1^{lim2}$ is a long and analytic expression which scales as $p_1^{\lim 2} = (eM)^{-1/2} + O(M^{-1})$. For $M > 4$, $p_1^{\lim 2} > p_1^{\lim 1}$. Furthermore, the maximal $p_1$ is given by $p_1^{\max} = (1 - 1/M)^{M-1}$. We conclude that there is a nontrivial interval $I = \left[ \max(p_1^{\lim 1}, p_1^{\lim 2}), p_1^{\max} \right]$ where we look for the minimal $p_2$. One can show that for large $M$, only the symmetric solution lies in $I$ (Supplementary Note 4). With a numerical study (an unconstrained maximization over $(a, b)$), we find that this happens when $M > 53$, but already for $M \gtrsim 30$, we observe that $p_1^{\max} - p_1^{\lim 2} \ll 1$ for all nonsymmetric configurations.

Numerically, we find that for all groups $M \geq 5$, the symmetric solution gives the global minimal $p_2$. From Eq. (11), we obtain $b^2 = p_1/Ma^{2-2M}$ and insert this into Eq. (12), which has to be minimized. With the parameters $p_1$ and $M$, this is a single-parameter polynomial and hence a numerically stable minimization is possible. The example $M = 3$ is discussed in the Supplementary Note 5 to demonstrate a case where a nonsymmetric configuration realizes the global minimum.

**Data availability**. All relevant data is available upon request.

Published online: xx xxx 2017

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

## Acknowledgements

We thank Philipp Treutlein, Otfried Gühne, Christoph Simon, Parisa Zarkeshian, Khabat Heshami, and Wolfgang Tittel for useful discussions. This work was supported by the European Research Council (ERC-AG MEC), the Swiss program National Centres of Competence in Research (NCCR) project Quantum Science Technology (QSIT) and the Swiss National Science Foundation (No. 200021_149109 and Starting grant DIAQ).

## Author contributions

F.F., N.B., F.B., and M.A. conceived the basic concept. F.F., C.G., N.B., and N.G. worked out the theory, supported by M.A. P.C.S., A.T., and J.L. performed the experiment and analyzed the data. F.F., A.T., N.B., and F.B. wrote the paper. All authors discussed the results and implications and commented on the manuscript at all stages.

## Additional information

**Competing interests:** The authors declare no competing financial interests.

