## [Peer Review File · Nature Communications]

Reviewers' Comments:

Reviewer #1:

None

Reviewer #3:

Remarks to the Author:

Since the authors did only stylistic alterations of the manuscript, my general view has not changed. In principle I support a publication back-to-back with the other manuscript in Nature Comm. - nevertheless the authors first have to address the questions raised below.

Here are some comments on the changes since the last version:

- 1) The introduction about entanglement depth has changed to less confusing, albeit still improvable.
- 2) The measurement procedure for p_1 has been clarified. Still, the question remains, how the retrieval of a single photon can be distinguished from other possibilities at the two detectors after the beamsplitter. The authors clarified, that the value for p_2 is inferred from a measurement of the autocorrelation of the photonic state right after the source, meaning before the storage. However, the detailed procedure is still not given. The inference of the atomic state of the memory from the photonic state before storage makes the results highly model dependent, which was suspected in the first review.
- 3) The number of atoms inferred from the doping concentration is now given. This estimate turned out to be a factor 7.5 bigger than the measured N.
- 4) Discussion: The authors now write: "Our results set a lower bound on the maximal number of entangled atoms allowed by any possible modification of quantum mechanics". If this statement is true, it needs to be clarified.
- 5) The authors now conclude with "We present numbers for the minimal necessary group size, which should not be confused with quantifying entanglement with an entanglement measure. Indeed, the nature of the target state, the W state D1, and the experimental challenges suggest that only a small amount of entanglement is present in the crystal during the storage."

At first sight, these statements seem to be in conflict with the main claims of the manuscript. The reason for this is, that there is no clear notion of the 'amount of entanglement' in a many-body situation. At present, this is mainly a matter of taste or possible application (for example metrologically useful entanglement is well defined). This should be clarified.

Reviewer #4:

Remarks to the Author:

Authors detect multipartite entanglement in solid state systems. With a single excitation, they create a W-state and detect it as multipartite entangled.

In particular, they create an atomic frequency comb and detect entanglement between particles. The

rare earth doped crystal absorbs a heralded single photon, then it emits it. Then they detect the photon and use the probability of having 1 and 2 photons to detect multipartite entanglement. They find large scale genuine multipartite entanglement of millions of particles. They find entanglement within each tooth.

I find the paper very well written, very clear. I would say, the paper describes an astonishing experimental success. I suggest its publication in Nature Communications.

I would like to react to the comments of the other referees.

I would now mostly comment concerning entanglement detection.

While the findings of the paper are relevant to applications, I would like to argue that multipartite entanglement in solids is in itself of a large importance.

— Why solid state systems are important: There has been a large effort to detect multipartite entanglement of many particles in quantum systems in large ensembles. These involved so far almost exclusively cold gases.

There has not been results in condensed matter systems. The reason is that in condensed matter systems there are many noise sources that do not exist in cold atoms. However, for the future of quantum information science, it is crucial to realize quantum information processing in condensed matter.

Just for this reason, the results of the paper are outstanding.

-- Why the detection of entanglement depth is important: most quantum systems can be considered to be in a product state of few particle states. Such as

$\Phi_1 \otimes \Phi_2 \otimes \Phi_3 \dots$

where Φ_n are states of at most k atoms. One can also consider mixture of such states, which are just the quantum states with an entanglement depth at most k .

Such states naturally appear, for example, in quantum states in thermal equilibrium that can very well be described by an ansatz

$\rho_1 \otimes \rho_2 \otimes \rho_3$

where ρ_k are units of at most k qubits. As the temperature increases, the minimal k that gives a good enough description of the state decreases.

Hence, the quantum system appears as being made of such groups of atoms that do not interact with each other.

Thus, in theory, the $T=0$ ground state of many spin models possess genuine multipartite entanglement, in practice, when such states are realized at finite temperatures and in a noisy environment, we can obtain quantum states that can be described by little particles groups among which there are only classical correlations.

Hence, the question arises: is it possible to have large scale entanglement in a quantum system? The answer could well be no. It could happen that they cannot create more than, say, 10 particle entanglement due to various noise effects.

On the other hand, the possibility of having large entanglement depth and large scale multipartite entanglement is necessary for real quantum information processing applications. And it is also a very important to prove that multipartite entanglement can be created from fundamental physics point of view.

Concerning future work, I would like to make a comment. One could try to find systems in which there is entanglement between spatially separated parties. Perhaps, this would be possible with some modification of the setup, for example, with two crystals. A proof of principle demonstration of entanglement between spatially separated parties could be very important and very interesting.

Reviewer #5:

None

Point-to-point answers to the referees (quotes from the reports in italic).

Referee #3:

1) The introduction about entanglement depth has changed to less confusing, albeit still improvable.

We are happy to read that the referee finds the introduction less confusing.

2) The measurement procedure for p_1 has been clarified. Still, the question remains, how the retrieval of a single photon can be distinguished from other possibilities at the two detectors after the beamsplitter. The authors clarified, that the value for p_2 is inferred from a measurement of the autocorrelation of the photonic state right after the source, meaning before the storage. However, the detailed procedure is still not given. The inference of the atomic state of the memory from the photonic state before storage makes the results highly model dependent, which was suspected in the first review.

The values for p_1 and p_2 are both obtained from the measurement statistics of the re-emitted light. This is clearly said in the introduction ("Then, we show that certain combinations of re-emission probabilities for one and two photons imply entanglement between a large number of atoms.") and in the caption of Fig. 1 (b) ("To estimate p_1 and p_2 , the one- and two-photon probabilities from the re-emission process are measured in the forward direction [...]"). Nevertheless, we improved the text in the mentioned caption as well as in the section Results to minimize any risk of confusion about this point.

This also implies that our claims are much less model-dependent as assumed by the referee. To make this clearer, we additionally added a paragraph in the section Discussion:

"As detailed in the Methods, our reasoning is based on two steps. First, a model-independent witness for entanglement depth is derived, which only depends on the overlap of the atomic state with $|D1\rangle$ and $|D2\rangle$ as well as the total number of atoms, N .

In the experiment, we measure the probabilities p_1 and p_2 for one and two photons, respectively, emitted from the atomic ensemble. The second step consists in identifying p_1 and p_2 with the probabilities of the atomic ensemble being in the $|D1\rangle$ and $|D2\rangle$ state before the re-emission, respectively. This step as well as the measurement of N are based on some assumptions regarding the atomic ensemble, the single-photon source and the light-matter interaction, Eq.(1). In addition, our claimed entanglement depth in the order of 10^7 takes finite detector efficiencies into account. We emphasize, however, that these assumptions have been thoroughly tested in the classical and quantum regime in many previous experiments. Further note that the entanglement depth is generated by a probabilistic but heralded source. Hence no post-selection has been made in our experiment."

3) The number of atoms inferred from the doping concentration is now given. This estimate turned out to be a factor 7.5 bigger than the measured N .

As mentioned in the Methods, the second method to estimate N is considered to be rather imprecise. A factor of 7.5 is hence not surprising. The number we use to calculate the entanglement depth comes from the method mentioned in the main text, which is more precise and leads to a smaller N (i.e., is more conservative).

4) Discussion: The authors now write: "Our results set a lower bound on the maximal number of entangled atoms allowed by any possible modification of quantum mechanics". If this statement is true, it needs to be clarified.

We agree that this statement was not well enough founded. We hence removed it.

5) *The authors now conclude with "We present numbers for the minimal necessary group size, which should not be confused with quantifying entanglement with an entanglement measure. Indeed, the nature of the target state, the W state D1, and the experimental challenges suggest that only a small amount of entanglement is present in the crystal during the storage."*

At first sight, these statements seem to be in conflict with the main claims of the manuscript. The reason for this is, that there is no clear notion of the 'amount of entanglement' in a many-body situation. At present, this is mainly a matter of taste or possible application (for example metrologically useful entanglement is well defined). This should be clarified.

We are a bit surprised that the referee had this impression as we never write about "amount of entanglement" in our manuscript. We consistently use the terminology "large number of entangled atoms", "large entanglement depth" or equivalent formulations. Nevertheless, we agree that it is necessary to carefully write and read papers from the field of "large entanglement", as there are many different facets of multipartite entanglement and a commonly accepted terminology has yet to be established.

Referee #4 did not ask for any changes of the manuscript. Regarding her/his suggestion of entanglement between spatially separated crystals, we refer to the paper Nature Photonics 6, 234–237 (doi:10.1038/nphoton.2012.34). This experiment is however not able to certify high entanglement depth (beyond two).

Reviewers' Comments:

Reviewer #3:

Remarks to the Author:

I am happy with the respond to my comments and support now publication in Nat. Comm.